# Uncover the Flavor Code of Roasted Sesame for Sesame Flavor Baijiu: Advance on the Revelation of Aroma Compounds in Sesame Flavor Baijiu by Means of Modern Separation Technology and Molecular Sensory Evaluation

**DOI:** 10.3390/foods11070998

**Published:** 2022-03-29

**Authors:** Hao Chen, Yashuai Wu, Junshan Wang, Jiaxin Hong, Wenjing Tian, Dongrui Zhao, Jinyuan Sun, Mingquan Huang, Hehe Li, Fuping Zheng, Baoguo Sun

**Affiliations:** 1Key Laboratory of Brewing Molecular Engineering of China Light Industry, Beijing Technology and Business University, Beijing 100048, China; ch15246300158@163.com (H.C.); wyss995418706@163.com (Y.W.); wangjs0816@163.com (J.W.); m15205199608@163.com (J.H.); sunjinyuan@btbu.edu.cn (J.S.); huangmq@th.btbu.edu.cn (M.H.); lihehe@btbu.edu.cn (H.L.); zhengfp@btbu.edu.cn (F.Z.); sunbg@btbu.edu.cn (B.S.); 2Beijing Laboratory of Food Quality and Safety, Beijing Technology and Business University, Beijing 100048, China; 3Department of Nutrition and Health, China Agriculture University, Beijing 100193, China; 4Department of Food and Bioengineering, Beijing Vocational College of Agriculture, Beijing 102442, China; 91505@bvca.edu.cn

**Keywords:** trace components, sesame flavor Baijiu, extraction method, detection technique, evaluation methods

## Abstract

Baijiu occupies an important position in the food industry of China and is deeply recognized as the national liquor of China. According to the flavor characteristics, Baijiu is artificially divided into 12 categories. The sesame flavor of Baijiu was accidentally discovered after the founding of the People’s Republic of China. Sesame flavor Baijiu is known for its special aroma of roasted sesame, which attracts people’s attention. Modern flavor extraction, separation technology, and flavor analysis technology, greatly promote the identification and evaluation of trace components and aroma compounds in Baijiu. Of note, it has successfully identified which aroma compounds are responsible for the special roasted sesame aroma in sesame flavor Baijiu. On this basis, this paper summarizes the extraction methods, detection techniques, analysis methods, aroma expression, and sensory evaluation methods that have been applied for the verification and evaluation of trace components and aroma compounds in Baijiu. More specifically, the research progress on the revelation of aroma compounds in sesame flavor Baijiu is systematically summarized. Next, people will focus on the changing mechanisms of aroma compounds and the metabolic regulation in Baijiu during brewing, which will be helpful for industrialization and the modern production of Baijiu.

## 1. Introduction

It was reported that the earliest alcoholic beverages were rice wine, fruit wine, and mead [1]. Afterwards, milk wine and grain wine gradually emerged, along with the development of society. Although the raw materials are quite different, wine is usually fermented through microorganisms using the sugars in raw materials, such as glucose, fructose in fruit, and lactose in milk. Although the sugars in the grain exist in the form of macromolecule polysaccharides (e.g., starch), during the brewing process of grain wine, polysaccharides are primarily converted into small molecule sugars by enzymatic hydrolysis, and then converted into ethanol by yeast [2].

Compared with Baijiu, Huangjiu appears earlier in China, and has a history of more than 7000 years [3]. Huangjiu is deemed as a kind of grain wine and is mainly fermented from rice and millet [1]. With the development of society and the emergence of distillation technology, Baijiu emerged. In contrast to the brewing process of fermented alcoholic beverages, the distillation process is added to the brewing process of Baijiu; therefore, Baijiu is a kind of distilled liquor with relatively high alcohol concentration.

So far, the jury is still out on when Baijiu originated and who created it. As early as the Tang dynasty, there were poems about Baijiu. At that time, Baijiu was named Shao Jiu or Shao Chun. At present, it is widely accepted that Baijiu has a history of more than 2000 years and can be traced back to the Western Han dynasty. In 2015, archaeologists discovered the Haihun Marquis Tomb in the Jiangxi Province of China. In the Haihun Marquis Tomb, experts found a bronze distiller in the liquor cellar. The bronze distiller is speculated to have been used for distilling liquor; hence, Baijiu is confirmed to have appeared during the Western Han dynasty. Huangjiu also has a history of more than 5000 years, and existed during the Western Han dynasty, thus, the material basis for distilling Baijiu is ample.

Liquor is the essence of food and is widely deemed as a precious item, as it has been used during important occasions since ancient times. Usually, three jin (a unit of weight, equal to 0.5 kg) of grain can steam out one jin of Baijiu; hence, rich grain is the material basis for brewing Baijiu. When society is prosperous and stable, and people live and work in peace and contentment, every household has surplus grain, and the surplus grain has a chance to be converted into liquor; therefore, liquor is often used as a treat for sacrifice among relatives and friends. In addition, the development of the liquor industry also partly reflects the level of national development.

Baijiu occupies an important position in the food industry of China and is deeply recognized as the national liquor of China. After the founding of China, with the prosperity of the country, the Baijiu industry developed in leaps and bounds; however, with the continuous development of the Baijiu industry, there have been continuous mergers among Baijiu companies, and the market has had a tendency to be saturated over the past five years (from 2016 to 2020). According to the data of the National Bureau of Statistics (as shown in Figure 1), the output of Baijiu has decreased, but profit is still increasing year on year, which implies that the Baijiu industry has gradually changed from pursuing output to quality.

According to the flavor characteristics, Baijiu is artificially divided into 12 categories, including strong, light, sauce, rice, mixed, feng, te, dong, chi, sesame, fuyu, and laobaigan flavored Baijiu [4]. The sesame flavored Baijiu was accidentally discovered after the founding of the People’s Republic of China [5]. The production technology of sesame flavor Baijiu combines the production technology characteristics of sauce flavor Baijiu, light flavor Baijiu, and strong flavor Baijiu. At present, the main producing areas of sesame flavor Baijiu are located in the Shandong, Henan, and Jiangsu provinces (as shown in Figure 2). More specifically, the production technology of sesame flavor Baijiu refers to the accumulation fermentation technology of strong flavor Baijiu, grain ripening technology of light flavor Baijiu, and cellar fermentation technology of strong flavor Baijiu.

In contrast to some distillates made from a single grain (e.g., whisky and vodka), Baijiu is made from a variety of grains (e.g., sorghum, wheat, rice husk, corn and other grains). In addition, Baijiu is obtained from a solid state fermentation process, and the process is relatively complex; thus, the trace components in Baijiu are more complex. Of note, the production technology of sesame flavor Baijiu (as shown in Figure 2) has been partially inherited from traditional technology. Baijiu has always been characterized by its open and natural multi-strain mixed fermentation [6]. Daqu is used as a matrix to naturally inoculate environmental microorganisms, which is applied to the fermentation of Baijiu [7]. Although the sesame flavor of Baijiu is based on the application of a series of optimized pure strains, namely, Bran Qu, the mixture of Daqu and Bran Qu is used as a saccharifying starter in sesame flavor Baijiu; hence, the main microorganisms and their metabolic pathways in the fermentation process of sesame flavor Baijiu are clearer and easier to control. In addition, the dependence of sesame flavor Baijiu quality on the environment is much lower than strong flavor and sauce flavor because the microorganisms involved in strong or sauce flavor Baijiu brewing are completely originate from the environment, whereas the microorganisms involved in sesame flavor Baijiu brewing are partly derived from pure strains [8]; therefore, the brewing process of sesame flavor Baijiu is more controllable, and the stability of the liquor quality is also improved.

As the name implies, sesame flavor Baijiu is known for its special roasted sesame aroma. Flavor quality is one of the most important factors that people consider when choosing Baijiu [9]. As reported, the flavor quality of Baijiu is mainly dependent on the trace components in Baijiu. Although water and ethanol are the main components of Baijiu, the difference in the types and concentrations of trace components in different Baijiu flavors are the key factors that cause a difference in Baijiu flavor quality [10].

In order to improve the flavor quality of Baijiu, the identification and quantification of trace components, as well as the screening and evaluation of aroma compounds, have become the focus of research on Baijiu. As a consequence, 2020 kinds of trace components have been found in Baijiu, including 510 esters, 249 alcohols, 140 acids, 18 lactones, 102 aldehydes, 160 ketones, 48 acetals, 82 sulfur compounds, 155 nitrogen compounds, 138 heterocyclic compounds, 170 aromatic compounds, 84 hydrocarbons, 104 terpenes, and 60 others [10]. Furthermore, a study on the trace components and aroma compounds of the sesame flavor of Baijiu has been carried out and has achieved crucial results. Moreover, this paper will note; 1. the extraction methods, detection techniques, and analysis methods for trace components in Baijiu, 2. the screening and evaluation methods of aroma compounds in Baijiu, 3. the research progress of trace components and aroma compounds in sesame flavor Baijiu, which were reviewed so as to provide guidance and a reference for the development of the Baijiu industry.

## 2. Introduction for the Extraction Method of Trace Components in Baijiu

As reported, the water solution of ethanol accounted for as much as 98% of Baijiu, whereas the trace components accounted for only 2% of Baijiu [10]. Although the concentration of trace components in Baijiu is extremely low, it is the trace components that play a crucial role in the formation of Baijiu flavor.

Therefore, the analysis on the trace components of Baijiu has been gradually developed. So far, 2020 kinds of trace components have been found in Baijiu [10]. Of note, extraction is the basic step to analyze the trace components in Baijiu.

At present, the extraction methods that have been applied for the extraction of trace components in Baijiu mainly include direct injection (DI) [11], liquid–liquid extraction (LLE) [12], solid phase microextraction (SPME) [13], stir bar sorptive extraction (SBSE) [14], and so on.

### 2.1. Direct Injection

Direct injection (DI) is a method whereby a sample is taken directly into analytical instruments for analysis. Direct injection is a method of direct separation by gas chromatography (GC). With this method, the sample is untreated or simply treated, so there is almost no sample loss prior to injection. As a result of its advantages, including simple operation, less loss, and high authenticity, direct injection has been widely used in the analysis of trace components in Baijiu. As early as 1994, researchers applied DI combined with a gas chromatography-flame photometric detector (GC–FPD) to analyze the trace components in sesame flavor Baijiu [11]. As a result, 50 kinds of trace components were found in sesame flavor Baijiu [15]. Afterwards, 68 kinds of trace components were detected in Baijiu using DI coupled with gas chromatography-mass spectrometry (GC–MS). In addition, more than 70 kinds of trace components were determined using DI combined with GC–MS [16]. Although DI is widely used, it has some limitations for the analysis of trace components in Baijiu. With DI, the sample is not enriched, thus, the enrichment rate of the sample is low. In fact, the concentrations of trace components in Baijiu are relatively low, thus, it is difficult to reach the detection limit of the instrument without enrichment. Therefore, when using DI combined with analytical instruments to detect the trace components in Baijiu samples, the kinds of trace components identified are relatively few; hence, the trace components in Baijiu cannot be comprehensively uncovered by only using DI.

### 2.2. Liquid–Liquid Extraction

Liquid–liquid extraction (LLE) applies the principle that target compounds have different solubilities in target solvents, which transfer target compounds from a sample matrix to the target solvent through multiple extractions. Compared with DI, LLE has a higher extraction efficiency but a more complex operation. Notably, since the extract obtained by LLE needs to go through a concentration step before injection, the enrichment rate of the sample is high and the enrichment capacity is strong. In addition, LLE can extract most of the compounds in Baijiu. As a consequence, LLE is widely used in the extraction and identification analysis of the trace components in Baijiu. In 2006, 85 kinds of trace components were detected by LLE combined with gas chromatography-olfactometry (GC–O) in Yanghe daqu (a kind of strong flavor Baijiu) [12]. In addition, LLE combined with GC–MS was used to extract and analyze vanillin, 4-methylguaiacol, and 4-ethylguaiacol from Gujinggong (a kind of strong flavor Baijiu) [17]. As a consequence, vanillin, 4-methylguaiacol, and 4-ethylguaiacol were present in all Gujinggong samples. Although LLE is widely used, it has its own limitations. In particular, LLE extracts trace components in Baijiu based on its polarity. During the extraction, semi-volatile and non-volatile components will be extracted along with volatile components. Thus, the existence of semi-volatile and non-volatile components will interfere with the separation and detection of volatile components, while in the pre-treatment process of LLE, the loss of volatile components will affect the detection and analysis of the trace components as well.

### 2.3. Solid Phase Microextraction

Solid phase microextraction (SPME) was proposed in 1989. This method is characterized by no solvent, a high degree of automation and good reproducibility; hence, SPME has a wide range of applications in the extraction and separation of trace components in Baijiu. In 2016, 1,1-diethoxymethane and methanethiol in sesame flavor Baijiu were identified and quantified using headspace-solid phase microextraction (HS–SPME) combined with GC–MS [13]; however, because of its thin coating, the enrichment ability is weaker than LLE. Furthermore, it is of note that although SPME is suitable for the analysis of various volatile and semi-volatile compounds in Baijiu, it also has its limitations due to the presence of non-volatile components in Baijiu.

### 2.4. Stir Bar Sorptive Extraction

Stir bar sorptive extraction (SBSE) is a novel pre-treatment method of solid phase extraction. Due to the larger volume of the stationary phase, the enrichment rate of SBSE is higher than that of SPME. In addition, it has the characteristics of a high extraction capacity, in that it has no need for an external agitator, a high degree of automation, etc.; therefore, this method has been widely used in the extraction of trace components of Baijiu. In 2011, the trace components in Baijiu were analyzed by SBSE coupled with GC–MS, and consequently, 76 kinds of trace components were verified [14]. Additionally, 18 kinds of pyrazines were extracted and identified in Jiannanchun (a kind of sauce flavor Baijiu) using SBSE combined with GC–MS [18]. Of note, there are few kinds of layers, and it is easy to absorb the non-volatile components, thereby affecting the extraction and identification of trace components in Baijiu.

### 2.5. Other Methods

In addition to the methods mentioned above, liquid–liquid microextraction (LLME) [19], simultaneous distillation extraction (SDE) [20], dynamic headspace (DHS) [21], and ultrasonic-assisted extraction (USAE) [5] have also been used to extract trace components in Baijiu. In 2013, a method of LLME was used to extract the trace components of Lang (a kind of sauce flavor Baijiu) and Fen (a kind of light flavor Baijiu) [19]. After analysis using GC–MS, 698 kinds of trace components were detected, including 167 esters, 67 alcohols, 11 lactones, 33 aldehydes, 48 kotenes, 18 acetals, 21 sulfur compounds, 66 nitrogen compounds, 11 heterocyclic compounds, 111 aromatic compounds, 8 hydrocarbons, 69 terpenes, 34 organic acids, 26 phenols, and 8 other kinds of trace compounds. In 2011, SDE was used to extract the trace components from Baijiu [22], and 8 kinds of trace components were identified by mass spectrometry. In general, these methods have their own characteristics and limitations in the process of extraction; therefore, a method cannot completely analyze all kinds of trace components in Baijiu. It is necessary to select specific methods according to the target ingredients and combine various methods to comprehensively analyze the trace components in Baijiu.

## 3. Introduction for the Detection Technique of Trace Components in Baijiu

With the advancement of science and technology, an increasing number of detection techniques have been applied for the analysis of trace components in Baijiu. At present, for the analysis of trace components in Baijiu, the most widely used detection techniques are GC–MS [23], the use of a gas chromatography-flame ionization detector (GC–FID) [24], a gas chromatography-sulfur chemiluminescence detector (GC–SCD) [25], a gas chromatography-flame photometric detector (GC–FPD) [26], a gas chromatography-nitrogen phosphorus detector (GC–NPD) [27], gas chromatography-olfactometry (GC–O) [28], and time of flight mass spectrometry (TOFMS) [25] and so on.

### 3.1. Mass Spectrometry

Mass spectrometry (MS) is an analytical method for measuring the mass–charge ratio of ions. The method relies on mass spectrometry for the qualitative analysis of trace components, and relies on the area of the chromatogram for quantitative analysis of trace components. MS is a kind of general detection technique for organic compounds, thus, it can pervasively detect trace components in Baijiu and has a wide range of uses. In 2012, LLE was used to separate the trace components in Tianzhilan (a kind of strong flavor Baijiu), and a total of 672 trace components were identified by GC–MS with a CP-Wax capillary column, including 26 kinds of sulfur compounds [23]. In 1992, 36 kinds of nitrogen compounds in mixed flavor Baijiu were detected by GC–MS [29]. Although MS is widely used in the qualitative and quantitative analysis of trace components in Baijiu, it also has the characteristics of poor specificity; therefore, it needs to combine multiple methods to analyze the trace components in Baijiu.

### 3.2. Flame Ionization Detector

A flame ionization detector (FID) is a kind of detector with a high sensitivity and low detection limit for the analysis of organic compounds. This method quantifies the trace components according to the intensity of the chromatographic peak, and it identifies the trace components according to the retention time; therefore, this method is not very accurate for the qualitative analysis of trace components and is generally used in combination with other detection techniques. Sulfur compounds in Baijiu were analyzed by GC combined with a FPD detector and FID detector [30], and three kinds of sulfur compound were identified and quantified. In 2014, the trace components in Gujinggong were extracted by LLE and were analyzed by GC–MS and a GC–FID [24]; consequently, 60 kinds of trace components were finally confirmed.

Thus, it is evident that a FID, as well as MS, are the most widely used detection techniques for an analysis of the trace components in Baijiu. At the same time, these two detectors are highly conventional detectors. As long as organic compounds are present, signals will be generated by the FID and MS for qualitative and quantitative analysis. In contrast, FIDs and MS have the characteristic of poor specificity. Baijiu contains a large number of alcohols, aldehydes, acids and esters, whose strong signals will cover the signals of characteristic compounds with a lower concentration (e.g., sulfur compounds and nitrogen compounds) in Baijiu, resulting in difficult identifications of sulfur and nitrogen compounds; therefore, specific detection techniques are also needed to detect the target trace components in Baijiu.

### 3.3. Sulfur Chemiluminescence Detector

A sulfur chemiluminescence detector (SCD) is a kind of specific detector with high sensitivity and selectivity for the detection of sulfur compounds. Sulfur compounds are identified as an important trace component in Baijiu. Although the concentration of sulfur compounds is extremely low in Baijiu, sulfur compounds make an important contribution to the flavor and quality of Baijiu, due to the fact that its aromatic characteristics are quite different from those of other major aroma compounds, and they have an extremely low odor threshold value. Thus, a SCD has been used to identify sulfur compounds in Baijiu. In 2020, characteristic sulfur compounds in Baijiu were analyzed by using LLE combined with GC × GC, equipped with a SCD; two sulfur compounds were identified as being representative of the aging process of Baijiu [25].

### 3.4. Flame Photometric Detector

Similar to the SCD, a flame photometric detector (FPD) is a kind of specific detector that identifies sulfur compounds in a sample. In general, a FPD is a detector that is highly selective and sensitive to sulfur compounds. Although sulfur compounds are verified as being crucial aroma compounds in Baijiu due to their characteristic aroma and extremely low odor threshold, the concentrations of sulfur compounds are much lower than those of other trace components, which greatly interferes with the analysis of sulfur compounds in Baijiu; therefore, as a detector for the identification of sulfur compounds, the use of a FPD greatly promotes the analysis of sulfur compounds in Baijiu. In 1993, Jin et al. extracted Baijiu samples with dichloromethane, and determined sulfur compounds in Baijiu by using a GC–FID and GC–FPD [31]. The result showed that 11 kinds of sulfur compounds were detected in Meilanchun (a kind of sesame flavor Baijiu). In 2014, the trace components in Baijiu were analysis by SPME and LLE [26]. Furthermore, a total of 13 sulfur compounds were quantified by GC–MS, a GC–FPD and GC–O.

### 3.5. Nitrogen Phosphorus Detector

Up until now, 155 nitrogen compounds had been found in Baijiu. Although there are fewer nitrogen compounds in Baijiu than other compounds, nitrogen compounds make a great contribution to the flavor of Baijiu, which is attributed to their special aromatic characteristics; therefore, increasing attention has been paid to the research on nitrogen compounds in Baijiu. Although a nitrogen phosphorus detector (NPD) provides technical support for the identification of nitrogen compounds in Baijiu, a NPD is a specific detector for the analysis of nitrogen compounds, and is widely used in the determination of nitrogen compounds in Baijiu. In 2014, 31 nitrogen compounds were detected in Baijiu samples from Guojing (a kind of sesame favor Baijiu) by LLE combined with GC–MS and GC–NPD [27]. More specifically, 23 nitrogen compounds were confirmed as pyrazines. Furthermore, 15 nitrogen compounds were found in Baijiu by HS–SPME combined with a GC–NPD [32]. Of note, 4-methylthiazole was first observed in Baijiu.

### 3.6. Gas Chromatography-Olfactometry

Gas chromatography-olfactometry (GC–O) separates and identifies potential aroma compounds from complex samples through the powerful separation capability of gas chromatography, and the sensitive human sense of smell. Different to the abovementioned detection techniques, GC–O combines compounds with aroma, thereby providing important technical support for flavor chemistry research. In 2015, 56 kinds of aroma compounds were found in chi flavor Baijiu using GC–O [33]. Furthermore, according to the result of odor activity values (OAVs), 34 aroma components were identified as the important aroma compounds of chi flavor Baijiu. In 2017, the potential aroma compounds were analyzed in Wuliangye (a kind of strong flavor Baijiu) using GC–O combined with AEDA [28]. As a consequence, a total of 62 aroma compounds were detected, and 45 aroma compounds were considered as being important aroma compounds of Wuliangye due to their higher flavor dilution (FD) factors.

### 3.7. Time of Flight Mass Spectrometry

Time of flight mass spectrometry (TOFMS) applies a precise mass number to characterize trace components. Different from quadrupole MS, TOFMS retains the complete molecular weight of the trace components. TOF can obtain the absolute value of the molecular weight for the unknown substance by taking advantage of the different falling time and moving time of molecules with different mass numbers in the electric field, and then it can deduce the molecular formula using the absolute value of molecular weight. TOFMS uses a high-resolution mass spectrometer that can accurately identify molecules, though its resolution, linearity, and stability are also good. It is often used in conjunction with two-dimensional gas chromatography (GC × GC). Based on GC × GC, trace components are separated using two columns. Of note, trace components that cannot normally be separated on the first column can be separated on the second column; therefore, the separation ability and degree of GC × GC are better than traditional one-dimensional chromatography, and it has been widely used in the analysis of trace components in Baijiu. In 2007, 528 trace components in Baijiu were detected by using GC × GC–TOFMS [34]. In 2017, the trace components in sesame flavor Baijiu were analyzed using GC × GC–TOFMS [35]; of note, 130 esters, 26 alcohols, 15 acids, 88 aldehydes and ketones, 16 nitrogen compounds, 20 furans, 14 terpenes, 25 sulfur compounds, and 6 other kinds of trace compound were detected. GC × GC can also be combined with other detectors; in 2020, GC × GC combined with a SCD was used to analyze the sulfur compounds in Baijiu. The result showed that 2-methyl-3-furanthiol and 2-furfurylthiol were detected [25].

## 4. Methods for the Analysis of Trace Components in Baijiu

The extraction and separation of trace components in Baijiu is the basis of Baijiu follow-up research, and the first step for the analysis on the trace components in Baijiu. On this basis, the identification analysis and concentration determination of trace components in Baijiu have been carried out. At present, the research methods that are used for qualitative analysis of trace components in Baijiu mainly include spectral library retrieval [36], standard comparison [32], retention index comparison [37], and aroma characteristic comparison [38]. The research methods that are used for the quantitative analysis of trace components in Baijiu mainly include the semi-quantitative method [39], the area normalization method [40], the external standard method [41], the internal standard method [42] and the stable isotope dilution method [43].

### 4.1. Qualitative Analysis of Volatile Substances in Baijiu

Qualitative analysis is the first step in recognizing the trace components in Baijiu, and it is the basis of subsequent research.

#### 4.1.1. Spectral Library Retrieval

Spectral library retrieval is the most basic method of qualitative analysis, and is widely used in the qualitative analysis of trace components in Baijiu. In detail, spectral library retrieval identifies the unknown compounds by comparing the mass spectra of unknown compounds in the sample with the mass spectra of the standard compounds in the spectral library. In 2016, Li et al. conducted qualitative analysis of trace components in Gujinggong using spectral library retrieval based on SPME combined with GC × GC–TOFMS [36]. The results showed that more than 1600 trace components were detected in Gujinggong, mainly consisting of alcohols and acids. However, spectral library retrieval is based on a mathematical similarity calculation, thus, different compounds with similar ion fragments are often misjudged as the same compound; hence, a manual resolution spectrum is necessary to improve the qualitative accuracy.

#### 4.1.2. Standard Comparison Method

The standard comparison method confirmed the unknown compound by comparing the spectral library retrieval with the retention time of the unknown compound in the sample and the standard in the working solution under the same monitored conditions. If the mass spectral and retention times of the unknown compound and the standard are consistent, the unknown compound is judged to be the known standard. Of note, the standard comparison method is widely used in the qualitative analysis of trace components in Baijiu because of its high accuracy. In 1992, sulfur compounds were determined in Baijiu by using the standard comparison method [44], and hydrogen sulfide and methyl sulfide were also verified. Dimethyl sulfur, dimethyl disulfide, and dimethyl trisulfide were determined in Baijiu by GC–MS combined with standard comparison [30]. In 2015, SPME combined with a GC–NPD was used to detect and analyze the trace components in Guojing [32]. Subsequently, 15 kinds of nitrogen compounds were identified by using the standard comparison method; however, the standard comparison method requires the purchase of an expensive standard, and not every standard can be bought, so this method also has certain limitations.

#### 4.1.3. Retention Index Comparison

The retention index comparison is a method to identify unknown compounds by comparing the retention index of an unknown compound and known compound. If their retention indices are similar, the unknown compound may be considered as the known compound. Generally, a retention index comparison does not require the purchase of a standard, and its accuracy is difficult to guarantee since it is greatly affected by instrument and environment conditions; therefore, it is usually used in combination with other methods as an auxiliary qualitative method. In 2014, LLE combined with GC–MS was used to analyze the trace components in Jingzhibaigan (a kind of sesame flavor Baijiu) and the trace components in Baijiu were qualitatively analyzed by NIST, 11 library searches, and a retention index comparison [45]. As a result, 65 kinds of trace components were identified, including 1 sulfur compound. In addition, DI and LLE combined with GC–MS was used to analyze the trace components in sesame flavor Baijiu [37], and then qualitative analysis was carried out by spectral library retrieval, a retention index comparison, and standard comparison.

#### 4.1.4. Aroma Characteristic Comparison

The aroma characteristic comparison is a method based on GC–O to assist the qualitative analysis of unknown compounds by comparing the aromatic characteristics of chromatographic effluents (i.e., unknown compounds) with known compounds. In general, this method has low accuracy and strong subjectivity, and the experimental group requires certain experience in sensory evaluation; therefore, it is often used in combination with other methods as an auxiliary means. LLE was used to extract the aroma compounds in Kouzijiao (a kind of mixed flavor Baijiu) and Jiannanchun [46]. The samples were separated and identified by GC–O and GC–MS. A total of 72 trace components were detected by LLE combined with GC–MS and GC–O in Maotai (a kind of sauce flavor Baijiu) [38]. The result showed that 3-methylbutanol, ethyl butanoate and ethyl hexanoate play an important role in the aroma of Baijiu.

### 4.2. Quantitative Analysis Method

Quantitative analysis is an analytical method to determine the concentration distribution of trace components in Baijiu based on the qualitative analysis of trace components.

#### 4.2.1. Semi-Quantitative Method

The semi-quantitative method is a method that applies only one reference object to quantitatively analyze the target compounds in samples. Thus, the operation is relatively simple and costs less. In 2019, a comparative method was established to analyze four kinds of sauce flavor Baijiu [39], and 31 kinds of organic acids were detected by GC–MS, and 25 kinds of organic acids were quantified by the semi-quantitative method. However, the response values of different compounds for the same detectors were different, thus, the result of the quantitative analysis has been largely deflected and the semi-quantitative method has a relatively low accuracy.

#### 4.2.2. Area Normalization Method

The area normalization method is based on the total peak area of the spectrum, and the percentage of the target compound relative to the reference is used to measure the concentration of the target compound. Although the procedure is simple, the area normalization method has the capacity for large errors and no definite quantitative results; therefore, it is generally used to roughly compare the relative concentration of trace components in Baijiu. In 2010, the area normalization method was used to quantitatively analyze the trace components in Baijiu [40]. The result showed that esters had a higher concentration in strong flavor Baijiu.

#### 4.2.3. External Standard Method

The external standard method is a widely used quantitative method for measuring the concentration of trace components in Baijiu. More specifically, a certain amount of a standard substance is added to the blank solvent, in accordance with the gradient to make a standard sample, and then the standard sample is processed and tested in conjunction with the unknown sample. The peak area of standard samples with different concentrations is used to draw a standard curve of response signal–concentration, so as to calculate the concentration of the target compounds (corresponding to the standard substance) in the unknown sample. Although the external standard method is more complex than the first two methods, the accuracy of the external standard method has been greatly improved. In 2015, Sun et al. used GC–MS/SIM combined with the external standard method for the quantitative analysis of 3-methylthiopropanol in sesame flavor Baijiu [41]. The result showed that the concentration of 3-methylthiopropanol ranged from 50 μg/L to 10 mg/L in sesame flavor Baijiu. Of note, the accuracy of the external standard method is affected by the stability of the instrument. Instrument fluctuation will reduce the accuracy of quantitative results.

#### 4.2.4. Internal Standard Method

The internal standard method adds the internal standard to the standard sample and unknown sample using the basis of the external standard method to eliminate accidental error and fluctuation of instrument noise, so as to improve the accuracy; therefore, it is a widely used method to determine the concentration of trace components in Baijiu. In 1997, GC–MS combined with the internal standard method was established to analyze the concentration of 68 kinds of trace components in Baijiu. The result showed that using three internal standards of quantitative determination is better than using a single internal standard [15]. In 2021, the concentration of ethyl acetate in Baijiu was determined using the internal standard method [42]. The result showed that the concentration of ethyl acetate was 2.421 g/L; however, the internal standard method is complex to operate. Of note, the selection of the internal standard is very critical, which directly affects the reliability of quantitative results.

#### 4.2.5. Stable Isotope Dilution Method

The stable isotope dilution method makes use of the principle that the chemical properties of isotope markers are the same as those of the substance to be measured, which eliminates the error in the whole extraction and separation process, and avoids accidental errors caused by the loss of trace components in the extraction process and the fluctuation of the instrument, thus improving the accuracy of quantification. In 2019, the stable isotope dilution method was used to quantify the concentration of phthalate esters in Baijiu, and the researchers performed a quantitative analysis in phthalate [43]. However, this method requires expensive markers or complex steps to synthesize the marker; thus, the stable isotope dilution method is normally applied to measure the crucial component in Baijiu.

## 5. Evaluation Methods for Aroma Compounds in Baijiu

Until now, 2020 kinds of trace components in Baijiu have been reported [10]. The question is whether so many trace components contribute directly to the flavor of Baijiu or not. The answer is no. As reported, only some of the trace components directly contribute to Baijiu flavor [10]; hence, screening aroma compounds from thousands of trace components has become a new research focus. Furthermore, relevant studies on the evaluation of aroma expression for aroma compounds have been gradually carried out. The widely applied evaluation methods were described as follows.

### 5.1. Frequency Method

The frequency method is a method based on GC–O with low requirements for sensory evaluators. This method simply involves determining whether sensory evaluators can smell a certain compound at a certain time. The level of aroma expression was evaluated based on the detection frequency of specific compound. Usually, the greater the detection frequency of a specific compound, the greater the contribution of the specific compound to the sample. This method has good reproducibility attributed to it, as it does not require evaluators to describe the aroma of a specific compound and it has low requirements for evaluators; therefore, it has been widely used to evaluate the aroma expression of aroma compounds in Baijiu. However, the frequency method could not evaluate the expression intensity of aroma compounds, thus, it could not uncover the relative importance of aroma compounds. Of note, due to the lack of effective information of aroma compounds, this method is usually used as an auxiliary evaluation method.

### 5.2. Time Intensity Method

Based on the frequency method, the time intensity method allows the sensory evaluators to score the expression intensity of each aroma compound; therefore, compared with the frequency method, the time intensity method has higher requirements for the sensory evaluators, but it still does not require the sensory evaluators to record aroma characteristics. GC–O was used to screen the aroma compounds in Yanghe Daqu, and the results showed that the aroma profile of Yanghe Daqu was mainly composed of esters and fatty acids. According to the time intensity method [12], ethyl hexanoate and ethyl butanoate were deemed to be the most important esters. However, the time intensity method is greatly affected by subjective factors, thus, its stability needs to be improved; therefore, some researchers have come up with other approaches, such as TDS (temporal dominance of sensations). The latest TDS claims to record multiple sensory attributes simultaneously over time, and the research based on TDS and TI have been carried out widely. In the future, it will provide a new impetus and lay a foundation for the sensory evaluation on Baijiu.

### 5.3. Dilution Method

Dilution is a common method for the evaluation of the expression intensity of aroma compounds based on GC-O, in which the sample is gradually diluted, and the sensory evaluation team records the time and properties of the aroma. The dilution method is based on the flavor dilution factor to evaluate the aroma expression of a target compound. The flavor dilution factor is the maximum dilution of a target compound that can still be perceived by humans. Relatively speaking, the dilution method can reduce the influence of subjective factors on the evaluation of results to a certain extent, and the flavor dilution factor is positively correlated with the aroma expression of target compound; hence, the reproducibility of dilution method is well established. Nowadays, it has been widely used to evaluate the expression intensity of aroma compounds in Baijiu. The most commonly used dilution method is the aroma extraction dilution analysis (AEDA). In 2016, a study was conducted on the evaluation of aroma compounds in Jingzhi (a kind of sesame flavor Baijiu) through HS–SPME–AEDA [47]. Consequently, 2-furfurylthiol, dimethyl trisulfide, *β*-damascenone, and methional were recognized as the characteristic aroma compounds of sesame flavor Baijiu. In 2017, the volatile compounds in Wuliangye were analyzed using GC–O combined with AEDA [28], and a total of 62 aroma compounds were detected. Based on a FD value, 45 aroma compounds were considered as being important aroma compounds. However, this method also has its limitations. The dilution method requires sensory evaluators to be trained and to have relevant knowledge of the aroma characteristics of aroma compounds in samples [48]. Moreover, the dilution method does not take into account the actual matrix effect in the sample, which reduces its practical value.

### 5.4. Odor Active Value

The frequency method, time intensity method, and dilution method are all based on GC–O to evaluate the aroma expression for the single aroma compound in the gas phase at different times, without considering the actual matrix effect. Although water and ethanol are the matrix of Baijiu, studies have shown that water and ethanol have important effects on the aroma expression of aroma compounds; therefore, in order to further improve the actual value of evaluation results, it is necessary to establish an evaluation method that considers the sample matrix effect. Moreover, an odor active value (OAV) was introduced to the evaluation on the aroma expression of aroma compounds in Baijiu. OAV is defined as the ratio of the concentration of the target aroma compound to the lowest concentration that can be perceived by the subject in a particular matrix. In fact, an aroma compound with a higher OAV is deemed to provide a greater contribution to the flavor of Baijiu. Generally, the aroma compound with an OAV greater than 1 is considered to contribute directly to the flavor formation of Baijiu. Nowadays, OAVs have been widely applied to the measurement of aroma compounds in Baijiu. In 2017, Zheng conducted the evaluation for the aroma compounds in sesame flavor Baijiu, and calculated the OAV of the aroma compounds [49]. As a result, the highest OAV value was obtained for ethyl hexanoate (OAV = 2691), followed by 3-methylbutanal (OAV = 2403), ethyl pentanoate (OAV = 1019) and ethyl octanoate (OAV = 782), which might strongly express the fruit and malt aromas. However, the OAV has its own limitations due to the fact that OAVs are based on the odor thresholds. Although the measurement process of the odor threshold is tedious, the deviation generated in the measurement process will affect the evaluation results.

### 5.5. Sensory Evaluation Method

Of note, all the above methods are aimed at evaluating the aroma expression of a single aroma compound, without considering the interaction effects between different aroma compounds in Baijiu; therefore, in order to further evaluate the contribution of aroma compounds to the Baijiu flavor, and to verify the reproducibility of Baijiu aroma profile by selected aroma compounds, sensory evaluation such as aroma recombination was introduced. Aroma recombination is based on an OAV to select target aroma compounds for recombination in order to evaluate the contribution of target aroma compounds to the aroma profile of Baijiu. Using a successfully established simulation system as a basis, the omission experiment was undertaken to further determine whether the absence of the selected aroma compound or aroma compounds of a certain category will cause a significant change to the overall aroma profile of Baijiu. The necessity of target aroma compounds for the formation of Baijiu flavor were then evaluated. In 2016, 56 kinds of trace components in Jingzhi were detected by GC–MS [50], then the experiment used GC–O combined with AEDA and an OAV to evaluate aroma compounds of sesame flavor Baijiu. In addition, the omission experiment also determined the important contribution of aroma compounds in sesame flavor Baijiu [49]. The result showed that ethyl hexanoate and methional played a key role in the aroma of sesame flavor Baijiu. In addition, QDA (quantitative descriptive analysis) was also commonly used in the sensory evaluation for Baijiu. QDA is a quantitative sensory analysis technique and was developed in the 1970s. It is a complete and accurate method for evaluators to assess the intensity of each flavor and note what comprises the aroma characteristics of samples. In 2019, Wang et al. applied QDA to describe the aroma profile of sesame flavor Baijiu. As a consequence, it was found that a sesame aroma and roasted aroma were significant in the aroma profile for sesame flavor Baijiu, followed by grain aroma [51].

## 6. Research Progress of Aroma Compounds in Sesame Flavor Baijiu

Baijiu is a traditional distilled liquor in China, which carries the experience and wisdom of Chinese people. Due to the differences in geographical environment, raw materials, brewing equipment, production technology, saccharification starter, storage and blending technique, Baijiu has obvious regional variety and flavor characteristics [52]. According to the flavor characteristics, Baijiu are roughly divided into 12 flavor types [28]. Sesame flavor Baijiu is a representative flavor type of Baijiu which was gradually formed through practice after the founding of China. The most notable feature of sesame flavor Baijiu is its characteristic aroma of roasted sesame, which is especially favored by consumers in the north. Moreover, sesame flavor Baijiu has formed certain production scales in the Shandong, Jiangsu, Henan provinces.

### 6.1. Qualitative Analysis of Trace Components in Sesame Flavor Baijiu

The formation of sesame flavor Baijiu was late, thus, the research on the trace components of sesame flavor Baijiu was carried out later than strong, light, and sauce flavor Baijiu. In 1957, sesame flavor was occasionally discovered in Jingzhibaigan [53]. In 1965, paper chromatography was used to analyze the trace components of Jingzhibaigan, which was the earliest exploration into sesame flavor Baijiu [54]. With the progress of science and technology, an increasing number of techniques and methods formed, which greatly promoted research on sesame flavor Baijiu. In 1991, the triangle theory was put forward by Hu [55], and it was established that the style and characteristics of sesame flavor Baijiu were different from strong, light, and sauce flavor Baijiu. In 1994, the research on the sulfur compound in sesame flavor Baijiu was carried out [11]. Consequently, 3-methylthiopropanol was identified by a GC–FPD combined with a standard comparison, and these were supposed to be the characteristic components of sesame flavor Baijiu. This study laid a foundation for the analysis of trace components in sesame flavor Baijiu, and the confirmation of sesame flavor Baijiu. Based on this research, in 1995, the former Light Industry Federation of China issued the industry standard of sesame flavor Baijiu (National Standard Number QB/T 2187-1995). Of note, 3-methylthiopropanol was identified as the characteristic component of sesame flavor Baijiu and the concentration of 3-methylthiopropanol in sesame flavor Baijiu was specified to be greater than or equal to 0.4 mg/L. With the establishment of industry standard, research on the analysis of trace components in sesame flavor Baijiu was gradually developed, which promoted the formation of the national standard for sesame flavor Baijiu. In 2006, the national standard of sesame flavor Baijiu (National Standard Number GB/T 20824-2007) [56] was officially promulgated, stipulating that the concentration of 3-methylthiopropanol in sesame flavor Baijiu should be no less than 0.5 mg/L for a high alcoholicity sample and no less than 0.4 mg/L for a low alcoholicity sample. Thus, sesame flavor Baijiu was widely recognized by Chinese people, which greatly promoted the development of research on its trace components.

In the inception phase, the study mainly focused on the qualitative analysis of trace components in sesame flavor Baijiu. In 2009, GC–FID was used to analyze the trace components of Shengliyuan (a kind of sesame flavor Baijiu) and 50 kinds of trace components were found by spectral library retrieval [57].

However, due to the limitations of techniques and methods, it was difficult to guarantee the accuracy of qualitative results and there were fewer trace components detected; thus, the trace components of sesame flavor Baijiu could not be fully explored. With the enrichment of techniques and the increase of personnel input, the number of trace components identified in sesame flavor Baijiu increased rapidly, and the accuracy was also greatly improved. In 2012, a total of 120 trace components were detected from two kinds of Meilanchun samples using HS–SPME combined with GC–MS. It includes 48 esters, 20 alcohols, 16 aldehydes and ketones, 15 organic acids, 3 alkanes, 2 phenols, 5 sulfur compounds, 8 nitrogen compounds, and 3 polyols [58]. In 2014, the trace components of Jingzhibaigan were analyzed using LLE combined with GC–MS [45]. The results showed that a total of 65 trace components were extracted with three kinds of solvents, including 9 alcohols, 19 esters, 11 acids, 14 hydrocarbons, 3 aromatic compounds, 3 furans, 3 aldehydes, 1 ketone, 1 kind of nitrogen compound and 1 sulfur compound. In 2017, 241 compounds were identified by DI and LLE in sesame flavor Baijiu, including 62 esters, 59 hydrocarbons, 24 acids, 32 alcohols, 10 furans, 8 aldehydes, 6 ketones, 25 aromatic compounds, 8 sulfur compounds, 6 nitrogen compounds, and 4 other compounds [49].

With the expansion of research, researchers found that during actual production, the quality of sesame flavor Baijiu fluctuated greatly. Although the sample of sesame flavor Baijiu reached the national standard, it did not have the typical roasted sesame aroma. It was speculated that 3-methylthiopropanol had no direct correlation with the typical roasted sesame aroma of sesame flavor Baijiu. Of note, the 3-methylthiopropanol had a typical meat and onion fragrance, which was not consistent with the characteristic aroma of sesame flavor Baijiu. Based on this, it was generally doubted whether the single compound of 3-methylthiopropanol could be identified as the characteristic component of sesame flavor Baijiu in the national standard of sesame flavor Baijiu (National Standard Number GB/T 20824-2007). This problem greatly restricted the development of sesame flavor Baijiu; therefore, the exploration of characteristic trace components in sesame flavor Baijiu has become the focus of research. Although the concentrations of sulfur compounds and nitrogen compounds in sesame flavor Baijiu is relatively low, both of them have a strong flavor releasing effect due to their extremely low odor threshold values [59], which are thought to play important roles in the formation of the characteristic flavor of Baijiu. Sesame flavor Baijiu has a special aroma of roasted sesame, similar to the flavor characteristics of coffee and sesame. Of note, sulfur compounds and nitrogen compounds exist in coffee and sesame seeds, and make an important contribution to the formation of coffee and roasted sesame aromas [60,61]; hence, sulfur and nitrogen compounds are presumed to be the characteristic components of sesame flavor Baijiu due to their similar aromas of roasted sesame. Thus, it is necessary to study sulfur and nitrogen compounds in sesame flavor Baijiu to reveal what contribution of sulfur and nitrogen compounds make to the formation of the characteristic flavor of sesame flavor Baijiu.

In 2012, LLE and SPME coupled with GC–MS was conducted to extract sulfur compounds in Guojing sesame flavor Baijiu, and 4 sulfur compounds were identified [62], including 3-methylthiopropanol, ethyl 3-methylthiopropionate, dimethyl trisulfide, and ethyl methylthioacetate. In 2014, a total of 31 nitrogen compounds were confirmed in sesame flavor Baijiu by LLE combined with GC–MS and a GC–NPD [27], including 14 pyrazine compounds, 1 pyrrole, 4 pyridines, 1thiazole, 1 oxazole and 2 other compounds. In 2016, SPME and LLE combined with GC–MS was used to conduct qualitative analysis on sulfur compounds in sesame flavor Baijiu [63]. Consequently, a total of 14 sulfur compounds were detected.

### 6.2. Quantitative Analysis of Trace Components in Sesame Flavor Baijiu

On the basis of the qualitative analysis for the trace components in sesame flavor Baijiu, in order to determine the concentrations of important trace components, researchers carried out quantitative analysis on the trace components in sesame flavor Baijiu. Researchers mainly focused on 3-methylthiopropanol as stipulated by the national standard, establishing corresponding quantitative analysis methods in accordance with the national standard. Thus, the concentration of 3-methylthiopropanol in sesame flavor Baijiu from different regions was determined. In 2014, a GC–FPD was used to monitor the concentration of 3-methylthiopropanol in sesame flavor Baijiu [64]. With the development of research, the sensitivity and accuracy of the method for the determination of 3-methylthiopropanol in sesame flavor Baijiu has been greatly improved. In 2015, a method for the determination of 3-methylthiopropanol in sesame flavor Baijiu by gas chromatography-mass spectrometry/selective ion scanning (GC–MS/SIM) combined with the external standard method was established [41]. The limit of detection was 5 µg/L and the limit of quantitation was 10 µg/L. Based on the results, it was also proven that 3-methylthiopropanol has no relationship with the typical sesame flavor of Baijiu.

Of note, not all base samples of sesame flavor Baijiu contained 3-methylthiopropanol. In addition, the concentrations of 3-methylthiopropanol in base samples were significantly different. The concentrations of 3-methylthiopropanol in some base samples were much lower than the concentration specified in the national standard. In addition, 3-methylthiopropanol also exists widely in chi flavor Baijiu and fruit wine. Among them, the concentration of 3-methylthiopropanol in sesame flavor Baijiu was similar to that in chi flavor Baijiu, but lower than that in fruit wine. This study further indicated that 3-methylthiopropanol was not the characteristic component of sesame flavor Baijiu. Recent studies have shown that there are many trace components in sesame flavor Baijiu, thus, the characteristic flavor of sesame flavor Baijiu may be formed by the interaction of different trace components. Based on the above results, it is not reasonable for the national standard of sesame flavor Baijiu (National Standard Number GB/T 20824-2007) to regard the single compound of 3-methylthiopropanol as the characteristic component of sesame flavor Baijiu. This research promotes the enhancement and revision of the national standard for sesame flavor Baijiu. Other important trace components in sesame flavor Baijiu were also quantitatively analyzed. In 2012, Meilanchun was analyzed by HS–SPME and GC–MS [65], which found that the concentration of tetramethyl pyrazine was 1.5~4.4 times higher than that of the same flavor Baijiu. In 2018, DI and GC–MS were used to analyze the other sulfur compounds in different brands of sesame flavor Baijiu [66]. Five sulfur compounds were detected and quantified by internal standard method.

### 6.3. Screening and Evaluation for the Aroma Compounds of Sesame Flavor Baijiu

In a further study, the researchers found that not all trace components were directly related to the flavor formation of sesame flavor Baijiu; hence, screening and evaluation of aroma compounds from thousands of trace components became a novel research focus.

In 2011, Fan et al. measured the olfactory thresholds of 79 trace compounds in Baijiu under the same substrate conditions [67]. It has promoted research investigating the expression of the aroma compounds.

In 2015, LLME and HS–SPME coupled with a GC–FID were used to screen and evaluate the aroma compounds in sesame flavor baijiu [68]. As a consequence, 87 aroma compounds in 36 kinds of sesame flavor baijiu with different brands, grades, and production years were quantitatively analyzed. Among them, 15 kinds of aroma compounds with an OAV greater than 100 were screened out and identified as important aroma compounds in sesame flavor baijiu, including 9 esters, 1 aldehyde, 1 alcohol, 2 sulfur compounds, 1 acid and 1acetal. Among them, esters mainly showed the fruit aroma, and pyrazines primarily contributed to the roasted aroma. Whereas sulfur compounds expressed the odor of rotten vegetables, alcohols mainly contributed to the aroma of grass. With the introduction of molecular sensory omics, the research on the necessity of aroma compounds for the aroma profile of sesame flavor baijiu has been gradually developed. In 2016, aroma compounds in Jingzhi were screened out and evaluated [50]. The aroma expression of 56 aroma compounds were further determined by AEDA combined with an OAV value calculation. The importance of ethyl hexanoate and methional to the overall aroma of sesame flavor baijiu was further confirmed by an omission experiment. On this basis, the aroma compounds of Jingzhi were further investigated by HS–SPME–AEDA [47]. As a result, the research showed that ethyl hexanoate, 2-furfurylthiol, dimethyl trisulfide, 3-methylbutanal, ethyl butanoate, ethyl 2-methylbutanoate, ethyl pentanoate, and ethyl 4-methylpentanoate were confirmed with higher FD factors. In particular, 8 sulfur compounds were identified as being potentially important to sesame flavor Baijiu. As a result, the concentration of these aroma compounds was further quantified by the internal standard method, then 36 aroma compounds were found to be above the corresponding odor threshold. On the basis of the sensory evaluation, 2-furfurylthiol (OAV = 1182), dimethyl trisulfide (OAV = 220), *β*-damascenone (OAV = 116), and methional (OAV = 99) could be the main source of the unique aroma of sesame flavor Baijiu.

In addition, the screening and evaluation of aroma compounds in other brands of sesame flavor Baijiu has also been gradually carried out. In 2018, 92 kinds of aroma compounds were detected in Meilanchun [69], among which 35 kinds of aroma compounds had OAVs ≥1, then an omission experiment showed that methional and dimethyl trisulfide played an important role to the flavor formation of sesame flavor Baijiu. After that, research was aimed to study the aroma compounds in Baotuquan (a kind of sesame flavor Baijiu) [70]. Consequently, 75 kinds of trace components in Baotuquan were detected by GC–MS; and finally, ethyl hexanoate, ethyl butanoate and 3-methylbutanal were further confirmed as important odorants after an omission experiment. In 2019, Li et al. explored the key aroma compounds in Guojing, and identified benzyl mercaptan for the first time [71], which makes an important contribution to the unique aroma of sesame flavor Baijiu. It had a strong roasted sesame aroma, thus, it was presumed to be the key aroma compound of sesame flavor baijiu.

So far, a total of 673 trace components have been identified in sesame flavor Baijiu, including 198 esters, 69 alcohols, 38 acids, 23 aldehydes, 30 ketones, 22 acetals, 73 nitrogen compounds, 61 sulfur compounds, 23 heterocyclic compounds, 52 aromatic compounds, 67 alkane compounds and 17 kinds of other compounds (as shown in Figure 3).

Among them, esters contribute to the fruity, flowery, and sweet aroma. Although sesame flavor Baijiu is characterized by its roasted sesame flavor, which is significantly different from other flavors of Baijiu, its main aroma compounds are still esters. Esters play an important role in the aroma profile for sesame flavor Baijiu. Of note, ethyl hexanoate (FD factor = 4096, OAV = 2691), ethyl butanoate (FD factor = 2048, OAV = 447), ethyl pentanoate (FD factor = 2048, OAV = 1019), and ethyl octanoate (FD factor = 25, OAV = 782), are the main aroma compounds of esters in sesame flavor Baijiu. Alcohols are the precursors of esters, and they mainly contribute to a fruity and fatty aroma. The common alcohols in sesame flavor Baijiu are 1-propanol (FD factor = 100, OAV = 6) and 2-methyl-1-propanol (FD factor = 8, OAV = 4). Acids contribute to a sour taste in the flavor profile of sesame flavor Baijiu, thus affecting the taste and the aftertaste of sesame flavor Baijiu. The lack of acids in sesame flavor Baijiu is the main reason for the poor aftertaste. Butanoic acid (FD factor = 1024, OAV = 57), pentanoic acid (FD factor = 256, OAV = 46), and hexanoic acid (FD factor = 2048, OAV = 35) are deemed to be the main acids in sesame flavor Baijiu. Aldehydes and ketones are also important aroma compounds in sesame flavor Baijiu and they contribute to a herbal and buttery aroma. For instance, 3-methylbutanal (FD factor = 8, OAV = 2403 and *β*-damascenone (OAV = 116) are recognized as important aroma compounds in sesame flavor Baijiu. Acetals such as 2-furaldehyde diethyl acetal (FD factor = 4) primarily contribute to a grassy, fruity, and sweet aroma. Heterocyclic compounds such as 2-acetyl furan (FD factor = 25) and 2-acetyl-5-methyl furan (FD factor = 5) mainly contribute to a nutty and sweet aroma in sesame flavor Baijiu. Phenolic compounds such as phenol (FD factor = 4), 4-methylphenol (FD factor = 1, OAV = 43), and 4-ethyl-2-methoxyphenol (FD factor = 4, OAV = 2) mainly contribute to a smoky aroma. Of note, the presumed reason why sesame flavor Baijiu differs from other flavor types of Baijiu is because of its sulfur compounds and nitrogen compounds (as shown in Figure 4). Sulfur compounds mainly expressed an aroma of roasted incense, scorched incense, coffee incense, whereas nitrogen compounds mainly presented nutty aromas, sweet aromas, and roasted aromas; therefore, sulfur compounds and nitrogen compounds are thought to be the characteristic aroma compounds of sesame flavor Baijiu.

Subsequently, 76 kinds of aroma compound were deemed to be responsible for the aroma profile of sesame flavor Baijiu through use of the aroma expression and static sensory evaluation method (as shown in Table 1). Of note, a total of 7 sulfur compounds and 11 nitrogen compounds made a significant contribution to the flavor characteristics of the roasted sesame aroma. More specifically, benzyl mercaptan exhibits a roasted taste. Dimethyl disulfide and dimethyl trisulfide exhibit an aroma of onion, cabbage, meat, and corn. Methional exhibits the aroma of roasted potato. Furfurylthiol exhibits the aroma of coffee, onion, roasted, sesame oil, and burnt incense. Difurfuryl ether also exhibits the aroma of coffee and roasted sesame. Ethyl 3-methylthio propionate exhibits a fruity aroma. Pyrazines (such as 2,6-dimethylpyrazine, 2-ethyl-6-methyl-pyrazine, 2-acetyl-pyrrole) primarily contribute to the nutty aroma.

At the same time, the multivariate analysis method has also been applied to the screening and evaluation of aroma compounds in sesame flavor Baijiu. In 2018, Sun et al. applied principal component analysis (PCA) to analyze the aroma compounds in sesame flavor Baijiu. The result showed that 6 compounds have higher PC1 and PC2 weight scores [66]. Indeed, the application of this method in this study on the aroma compounds in Baijiu also has broad prospects for development.

## 7. Conclusions

So far, a total of 673 trace components have been identified in sesame flavor Baijiu, including 198 esters, 69 alcohols, 38 acids, 23 aldehydes, 30 kotenes, 22 acetal, 73 nitrogen compounds, 61 sulfur compounds, 23 heterocyclic compounds, 20 phenols, 32 aromatic compounds, 67 hydrocarbon alkyl, and 17 kinds of other compounds. Of note, the reason why sesame flavor Baijiu differs from other flavors of Baijiu is because of its sulfur compounds and nitrogen compounds. Molecular sensory omics indicate that a total of 7 sulfur compounds and 11 nitrogen compounds make a significant contribution to the special roasted sesame aroma of sesame flavor Baijiu.

Based on the above research, exploring key aroma compounds and the interactions among these key aroma compounds, and investigating the change mechanism of the aroma compounds in the production process, will help improve the production technology for sesame flavor Baijiu, strengthening the aroma characteristic of sesame flavor Baijiu, and improving the quality of sesame flavor Baijiu.

## Figures and Tables

**Figure 1 foods-11-00998-f001:**
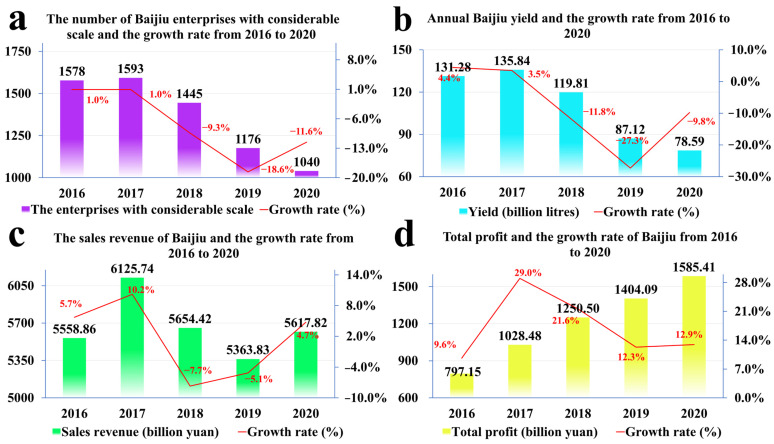
Overview of the Baijiu industry from 2016 to 2020.

**Figure 2 foods-11-00998-f002:**
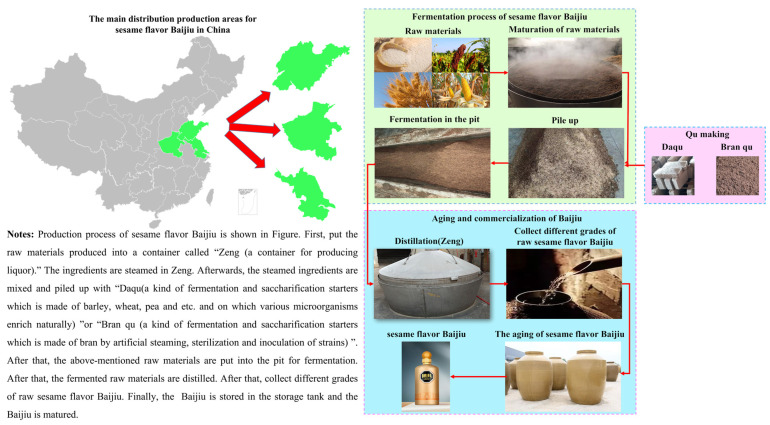
Production process of sesame flavor Baijiu.

**Figure 3 foods-11-00998-f003:**
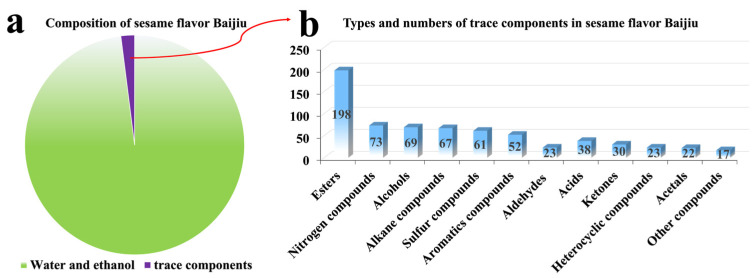
Types and numbers of trace components in sesame flavor Baijiu.

**Figure 4 foods-11-00998-f004:**
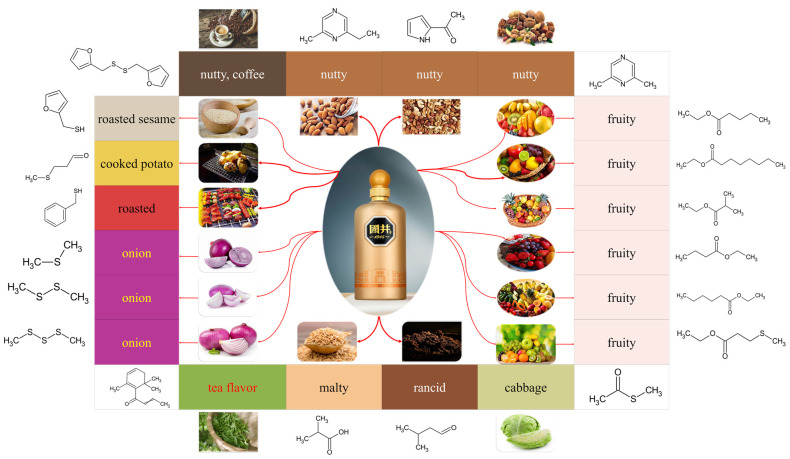
The key aroma components for the characteristic aroma of sesame flavor Baijiu.

**Table 1 foods-11-00998-t001:** The important aroma components for sesame flavor Baijiu.

No.	Aroma Compounds	CAS Number	Aroma Descriptors	FD Factor	OAV	Ref. ^1^ (Year)
1	ethyl hexanoate	123-66-0	fruity	4096	2691	[50] (2016)
2	3-methylbutanal	590-86-3	malty	8	2403	[50] (2016)
3	ethyl acrylate	140-88-5	plastic	512	2225	[50] (2016)
4	2-furfurylthiol	98-02-2	roasted sesame seeds	400	1182	[47] (2017)
5	ethyl pentanoate	539-82-2	fruity	2048	1019	[50] (2016)
6	ethyl octanoate	106-32-1	fruity	25	782	[50] (2016)
7	ethyl 2-methylpropanoate	97-62-1	fruity, sweet	100	600	[47] (2017)
8	3-methylbutyl acetate	123-92-2	fruity		495	[47] (2017)
9	ethyl butanoate	105-54-4	fruity	2048	447	[50] (2016)
10	dimethyl trisulfide	3658-80-8	onion	128	388	[41] (2016)
11	benzyl mercaptan	100-53-8	roasted	19,683	286	[64] (2019)
12	2-methylpropanoic acid	79-31-2	rancid	4	226	[47] (2017)
13	ethyl 2-methylbutanoate	7452-79-1	fruity	400	215	[50] (2016)
14	*β*-damascenone	23696-85-7	tea flavor		116	[47] (2017)
15	3-methylbutanoic acid	503-74-2	sweaty	2048	89	[50] (2016)
16	butanoic acid	107-92-6	sweaty, rancid	1024	57	[50] (2016)
17	ethyl acetate	141-78-6	pineapple	100	56	[47] (2017)
18	ethyl decanoate	110-38-3	fruity	16	46	[50] (2016)
19	pentanoic acid	109-52-4	sweaty	256	46	[50] (2016)
20	hexanoic acid	142-62-1	sweaty	2048	35	[50] (2016)
21	geosmin	104873-46-3			22	[47] (2017)
22	methional	3268-49-3	cooked potato	25	99	[50] (2016)
23	ethyl lactate	97-64-3	fruity	16	15	[50] (2016)
24	dimethyl sulfide	75-18-3	cooked onion	5	14	[47] (2017)
25	s-methyl thioacetate	1534-08-3	rotten cabbage	100	13	[47] (2017)
26	2-phenylethyl acetate	103-45-7	floral	32	7	[50] (2016)
27	1-propanol	71-23-8	fruity	100	6	[47] (2017)
28	2-methyl-1-propanol	78-83-1	fruity	8	4	[50] (2016)
29	ethyl 3-phenylpropanoate	2021-28-5	floral	16	3	[50] (2016)
30	phenylacetic acid	103-82-2	sweet, honey	2	3	[50] (2016)
31	ethyl phenylacetate	101-97-3	fruity, sweet	64	3	[50] (2016)
32	4-ethyl-2-methoxyphenol	2785-89-9	smoky	4	2	[50] (2016)
33	3-methyl-1-butanol	123-51-3	malty	1024	2	[50] (2016)
34	2-furfural	98-01-1	sweet	64	2	[50] (2016)
35	furfuryl alcohol	98-00-0	caramel	16	1	[50] (2016)
36	ethyl benzoate	93-89-0	fruity	32	<1	[50] (2016)
37	acetic acid	64-19-7	vinegar	5	<1	[50] (2016)
38	5-methyl furfural	620-02-0	baked	25	<1	[50] (2016)
39	3-methylthio-1-propanol	505-10-2	cooked vegetable	100	<1	[47] (2017)
40	trimethyl pyrazine	14667-55-1	nutty	4	<1	[50] (2016)
41	octanoic acid	124-07-2	sweaty	32	<1	[50] (2016)
42	diethyl butanedioate	123-25-1	sweet	25	<1	[50] (2016)
43	2-acetyl furan	1192-62-7	sweet	25	<1	[50] (2016)
44	heptanoic acid	111-14-8	sweaty	32	<1	[50] (2016)
45	diethyl acetal	105-57-7	fruity	512	<1	[50] (2016)
46	ethyl propanoate	105-37-3	fruity	100	<1	[50] (2016)
47	benzyl alcohol	100-51-6	sweet	16	<1	[50] (2016)
48	propanoic acid	79-09-4	sour	128	<1	[50] (2016)
49	2-phenylethanol	60-12-8	rose-like	256	<1	[50] (2016)
50	ethyl 3-methylthio propionate	13327-56-5	fruity		<1	[67] (2016)
51	terpineol	8000-41-7	floral	200		[47] (2017)
52	hexyl hexanoate	6378-65-0	apple, peach	10		[47] (2017)
53	propyl hexanoate	626-77-7	fruity	200		[47] (2017)
54	ethyl 2-mercaptoacetate	623-51-8	cooked vegetable	100		[47] (2017)
55	1-octen-3-one	4312-99-6	mushroom	5		[47] (2017)
56	ethyl benzoate	32874-26-3	fruity	10		[50] (2016)
57	3-methylbutyl hexanoate	2198-61-0	fruity	100		[47] (2017)
58	naphthalene	16728-99-7	musty	5		[70] (2019) [47] (2017)
59	benzaldehyde	100-52-7	fruity	5		[70] (2019) [50] (2017)
60	ethyl 3-(methylthio) propanoate	13532-18-8	sulfur, rotten cabbage	5		[70] (2019) [47] (2017)
61	2,5-dimethyl-3-ethylpyrazine	13360-65-1	baked	100		[70] (2019) [47] (2017)
62	2-acetyl-5-methyl furan	1193-79-9	baked	5		[69] (2018)
63	1-hexanol	111-27-3	floral	25		[47] (2017)
64	2,6-dimethylpyrazine	108-50-9	nutty	100		[69] (2018) [47] (2017)
65	methyl hexanoate	106-70-7	floral	25		[69] (2018)
66	benzoic acid	65-85-0	balsam	2		[50] (2016)
67	ethyl hexadecanoate	628-97-7	waxy	1		[50] (2016)
68	ethyl 2-hydroxybutanoate	52089-54-0	fruity	4		[50] (2016)
69	ethyl 4-methylpentanoate	25415-67-2	fruity	16		[50] (2016)
70	2-furaldehyde diethyl acetal	13529-27-6	sweet	4		[50] (2016)
71	phenol	108-95-2	phenolic	4		[50] (2016)
72	ethyl 3-methylbutanoate	108-64-5	berry	512		[50] (2016)
73	2-ethyl-6-methyl-pyrazine	13925-03-6	nutty	32		[50] (2016)
74	difurfuryldisulfide	224-649-6	nutty, coffee	4		[50] (2016)
75	2-acetyl-pyrrole	214-016-2	nutty	3		[50] (2018)
76	dimethyl disulfide	624-92-0	onion, cabbage, meat, corn.	3		[50] (2018)

^1^ Data from the following references.

## Data Availability

Not applicable.

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
