# Peer review of "Uncover the Flavor Code of Roasted Sesame for Sesame Flavor Baijiu: Advance on the Revelation of Aroma Compounds in Sesame Flavor Baijiu by Means of Modern Separation Technology and Molecular Sensory Evaluation"

_foods, 2022, doi:10.3390/foods11070998_

Round 1

Reviewer 1 Report

Making a comparison with a distillate from very similar raw materials would be very interesting to highlight the potential of Baijiu.

Lines 86 to 95 unnecessary history.
Lines 496 to 512 unclear method.
The part concerning sensory analysis is to be explored. 

Author Response

Response to Reviewer 1 Comments

Thank you very much for your comments concerning our manuscript entitled “Uncover the flavor code of roasted sesame for sesame flavor Baijiu: Advance on the revelation of aroma compounds in sesame flavor baijiu by means of modern separation technology and molecular sensory evaluation” (ID: foods-1611064). Those comments are all valuable and very helpful for revising and improving our paper. We have studied comments carefully and have made correction which we hope meet with approval. Revised portions were marked by blue in "Tracked Changes" mode in our revised version. Other changes are also highlighted in bule. The main corrections in the paper and the responds to the reviewer’s comments are as following:

Point 1: Making a comparison with a distillate from very similar raw materials would be very interesting to highlight the potential of Baijiu.

Response 1: Thanks very much for your approval of this review and thank you for your patient review and kind advices. According to your suggestion, we have made a comparison with distillates from very similar raw materials on line 89 to 93. In detail, "In addition, the production technology of sesame flavor Baijiu (as shown in Figure 2) is the inheritance and development of traditional technology." was reversed as "In contrast to some distillates made from single grain (e.g., whisky and vodka), Baijiu is made from a variety of grains (e.g., sorghum, wheat, rice husk, corn and other grains). In addition, Baijiu is obtained from a solid state fermentation process, and the process is relatively complex. Hence, the trace components in Baijiu are more complex. Of note, the production technology of sesame flavor Baijiu (as shown in Figure 2) is the inheritance and development of traditional technology." The complex process of Baijiu was also introduced in the following part. Thank you again for your kind review.

Point 2: Lines 86 to 95 unnecessary history.

Response 2: Thank you for your patient reading and kind advices. It is true that some of the concepts in this section are not concise enough, so we have revised this section and added a comparison between Baijiu and other distillates on line 89 to 93. In detail, "In addition, the production technology of sesame flavor Baijiu (as shown in Figure 2) is the inheritance and development of traditional technology." was reversed as "In contrast to some distillates made from single grain (e.g., whisky and vodka), Baijiu is made from a variety of grains (e.g., sorghum, wheat, rice husk, corn and other grains). In addition, Baijiu is obtained from a solid state fermentation process, and the process is relatively complex. Hence, the trace components in Baijiu are more complex. Of note, the production technology of sesame flavor Baijiu (as shown in Figure 2) is the inheritance and development of traditional technology." Thank you again for your careful review.

Point 3: Lines 496 to 512 unclear method.

Response 3: Many thanks for your patient guidance. Our method description in this section needs to be explained in more depth. According to your suggestion, we have added some details and elaborated the method on line 505 to 512. In detail, " Relatively, dilution method can reduce the influence of subjective factors on the evaluation results to a certain extent, and the dilution factor is used to express the aroma expression intensity of aroma compounds." was reversed as "Dilution is a common method for the evaluation on the expression intensity of aroma compounds based on GC-O, in which the sample is gradually diluted, and the sensory evaluation team records the time and properties of the aroma. The dilution method is based on the flavor dilution factor to evaluate the aroma expression of target compound. Flavor dilution factor means the maximum dilution of a target compound that can be perceived by the human sense. Relatively, dilution method can reduce the influence of subjective factors on the evaluation results to a certain extent, and the flavor dilution factor is positively correlated with the aroma expression of target compound." Many thanks again for your review.

Point 4: The part concerning sensory analysis is to be explored.

Response 4: Thank you for your careful review and helpful guidance. We have modified the part concerning sensory analysis according to your requirements. We have described the characteristics and representative aroma compounds of each category of compounds in detail and refined this part on line 760 to 788. In detail, " Among them, esters contribute the aroma of fruit, flower and sweet. Alcohols main represent the aroma of fruit and fatty. Acids show the aroma of sour. Aldehydes and ketones show the aroma of herbal and buttery. Acetals mainly show the aroma of grass, fruit and sweet. Heterocyclic compounds mainly show nutty aroma and sweet aroma. Phenolic compounds mainly exhibit smoky aroma. Of note, the reason why sesame flavor Baijiu differs from other flavor types of Baijiu is presumed to be attributed to its sulfur compounds and nitrogen compounds (as shown in Figure 4). Sulfur compounds mainly expressed roasted incense, scorched incense, coffee incense, and nitrogen compounds mainly present nutty aroma, sweet aroma and roasted aroma. Therefore, sulfur compounds and nitrogen compounds are supposed to be the characteristic aroma compounds of sesame flavor Baijiu." was reversed as "Among them, esters contribute the aroma of fruit, flower and sweet. Although sesame flavor Baijiu is characterized by roasted sesame flavor, which is significantly different from other flavor Baijiu, its main aroma compounds is still esters. Esters play an important role in the aroma profile for sesame flavor Baijiu. Of note, ethyl hexanoate (FD factor = 4096, OAV = 2691), ethyl butanoate (FD factor = 2048, OAV = 447), ethyl pentanoate (FD factor = 2048, OAV = 1019), ethyl octanoate (FD factor = 25, OAV = 782) are the main aroma compounds of esters in sesame flavor Baijiu. Alcohols are the precursors of esters, it main represents the aroma of fruit and fatty. For instance, 1-propanol (FD factor = 100, OAV = 6), 2-methyl-1-propanol (FD factor = 8, OAV = 4) are the common alcohols in sesame flavor Baijiu. Acids contribute sour taste for flavor profile of sesame flavor Baijiu and affect the taste and the aftertaste of sesame flavor Baijiu. The lack of acids in sesame flavor Baijiu is the main reason for the poor aftertaste. Butanoic acid (FD factor = 1024, OAV = 57), pentanoic acid (FD factor = 256, OAV = 46), and hexanoic acid (FD factor = 2048, OAV = 35) are deemed as the main acids in sesame flavor Baijiu. Aldehydes and ketones are also important aroma compounds in sesame flavor Baijiu and they show the aroma of herbal and buttery. For instance, 3-methylbutanal (FD factor = 8, OAV = 2403 and β-damascenone (OAV = 116) are recognized as important aroma compounds in sesame flavor Baijiu. Acetals primary show the aroma of grass, fruit and sweet, such as 2-furaldehyde diethyl acetal (FD factor = 4). Heterocyclic compounds mainly contribute nutty aroma and sweet aroma for sesame flavor Baijiu, such as 2-acetyl furan (FD factor = 25) and 2-acetyl-5-methyl furan (FD factor = 5). Phenolic compounds mainly exhibit smoky aroma, such as phenol (FD factor = 4), 4-methylphenol (FD factor = 1, OAV = 43) and 4-ethyl-2-methoxyphenol (FD factor = 4, OAV = 2). Of note, the reason why sesame flavor Baijiu differs from other flavor types of Baijiu is presumed to be attributed to its sulfur compounds and nitrogen compounds (as shown in Figure 4). Sulfur compounds mainly expressed roasted incense, scorched incense, coffee incense, and nitrogen compounds mainly present nutty aroma, sweet aroma and roasted aroma. Therefore, sulfur compounds and nitrogen compounds are supposed to be the characteristic aroma compounds of sesame flavor Baijiu." Thank you again for your careful review.

We tried our best to improve the manuscript and made some other changes in the manuscript. These changes will not influence the content and framework of the paper. And here we did not list the changes but marked by red in revised version. We appreciate the reviewers’ warm work earnestly, and hope that the correction will meet with approval. If there are any problems about our paper, please do not hesitate to let us know.

Dear editor,

We will revise the manuscript as soon as possible.

Thank you for your letter

Thank you and best regards,

Yours sincerely,

Dongrui Zhao

Key Laboratory of Brewing Molecular Engineering of China Light Industry

Beijing Laboratory for Food Quality and Safety

Beijing Technology and Business University, Beijing, China 100048

Phone: 86-10-68984890. Fax: 86-10-68984890

E-mail: zdrui6789@sina.com

Reviewer 2 Report

Section 1-5 reads fine where all information is subheaded accordingly and was easy to read.

Section 6 needs more structure - it is quite difficult to read at the current stage. Some form of subheaders is needed.

Figure 4 is concerning, fruity for example refers to multiple fruits - or another as pineapple. Can the authors clarify this?

What would be interesting is for the authors to compare sensory methods applied for this. Was there any other studies that attempted QDA and link the OACs to sensory attributes? 

Some information on the techniques applied - e.g. PCAs PLSRs would be beneficial for the readers as well.

Isn't TI Section 5.2 a sensory technique? Shouldn't it be parked under 5.5? A quick search on Scholar also showed the combined use of TI and TDS which isn't reviewed here. 

I'd recommend the authors to expand the sensory section as it is rather lacking while the other sections is already fine.

Author Response

Dear Editor and Reviewer,
Thank you very much for your comments concerning our manuscript entitled “Uncover the flavor code of roasted sesame for sesame flavor Baijiu: Advance on the revelation of aroma compounds in sesame flavor baijiu by means of modern separation technology and molecular sensory evaluation”
(ID: foods-1611064). Those comments are all valuable and very helpful for revising and improving our paper. We have studied comments carefully and have made correction which we hope meet with approval. Revised portions were marked by red in "Tracked Changes" mode in our revised version. Other changes are also highlighted in red. The main corrections in the paper and the
responds to the reviewer’s comments are as following:

Point 1: Section 1-5 reads fine where all information is subheaded accordingly and was easy to read.

Response 1: It's our pleasure to invite you to review this paper and thank you very much for your careful review concerning our manuscript. We are greatly encouraged by your recognition for section 1-5. Thank you again for your careful review.

Point 2: Section 6 needs more structure - it is quite difficult to read at the current stage. Some form of subheaders is needed.

Response 2: Thank you for your patient reading and kind advices. This part is indeed a little hard to read and needs more structure. Therefore, we have redrawn the structure of section 6 and we have added 3 subtitles on line 587, 669 and 707 in "Tracked Changes" mode. At the beginning of the section 6, the general situation of sesame flavor Baijiu was summarized. In 6.1., qualitative analysis of trace components in sesame flavor Baijiu was carried out, in 6.2., quantitative analysis of sesame flavor Baijiu was carried out, and in 6.3., screening and evaluation of aroma compounds in sesame flavor Baijiu were carried out. Thank you again for your review.

Point 3: Figure 4 is concerning, fruity for example refers to multiple fruits - or another as pineapple. Can the authors clarify this?

Response 3: Many thanks for your patient guidance. This part of the aroma compounds reflect a general fruit note, does not refer to a particular fruit. It is very sorry that the Figure 4 did not express clearly. Thank you for your careful review and patient guidance, we have modified Figure 4 according to your request and changed Figure 4 of the specific fruit into the overall description for fruit, thank you again for your guidance.

Point 4: What would be interesting is for the authors to compare sensory methods applied for this. Was there any other studies that attempted QDA and link the OACs to sensory attributes?

Response 4: Thank you very much for your careful review. According to your suggestions, we have explained the concept of QDA that link the OACs to sensory attributes. Moreover, we have conducted literature research and supplemented relevant references on line 568 to 575. In detail, "In addition, QDA (quantitative descriptive analysis) is also commonly used in the sensory evaluation for Baijiu. QDA is a quantitative sensory analysis technique and developed in the 1970s. It is a complete and accurate method for evaluators to assess the intensity of each flavor note that constitutes the aroma characteristics of samples. In 2019, Wang et al. applied QDA to describe the aroma profile of sesame flavor Baijiu. As a consequence, it was found that sesame aroma and baking aroma were significant in the aroma profile for sesame flavor Baijiu, followed by grain aroma [51]" has been added at the end of section 5.5. Thanks again for your review.

Point 5: Some information on the techniques applied - e.g. PCAs PLSRs would be beneficial for the readers as well.

Response 5: Thank you for your careful review. In this paper, we focus on the front-end of some qualitative and quantitative methods. Moreover, sensory evaluation methods were described. For the multivariate statistical analysis applied in the analysis for the aroma compounds in Baijiu is not described as the focus of this paper, but the relevant research on this part has already been carried out, and certain results have been achieved, so we have re-examined the literature and made a related discussion to this part at the end of section 5 on line 804 to 809. Besides, we have added "At the same time, multivariate analysis method has also been applied in the screening and evaluation of aroma compounds in sesame flavor Baijiu. In 2018, Sun et al. applied principal component analysis (PCA) to analysis the aroma compounds in sesame flavor Baijiu. The result showed that 6 compounds have higher PC1 and PC2 weight scores [66]. Indeed, the application of this method in the study on the aroma compounds in Baijiu also has broad prospects for development." at the end of section 5.5. Many thanks again for your patient review.

Point 6: Isn't TI Section 5.2 a sensory technique? Shouldn't it be parked under 5.5? A quick search on Scholar also showed the combined use of TI and TDS which isn't reviewed here.

Response 6: We are very grateful to your comments for the manuscript. Thank you for your patient review. The time intensity method, similar to frequency method and dilution method, is a preliminary screening method for aroma expression intensity of aroma compounds in the gas phase medium. This method is considered as a relatively simple screening method for aroma compounds, the matrix effect (mainly ethanol) is not considered, so it is juxtaposed with frequency method and dilution method. Based on, time intensity method, frequency method and dilution method are aimed at evaluating the aroma expression of a single aroma compound without considering the interaction effect between different aroma compounds in Baijiu. Therefore, in order to further evaluate the contribution of aroma compounds to Baijiu flavor and verify the reproducibility of Baijiu aroma profile by selected aroma compounds, sensory evaluation such as aroma recombination was introduced. Based on the above methods, sensory evaluation consider the actual matrix effect the interaction effect between different aroma compounds in Baijiu and the interaction effect between different aroma compounds in Baijiu, so the article puts this approach behind the previous methods.

After your suggestion, we have conducted a survey on TDS, which is indeed a widely used method in study on aroma compounds, but less used in the analysis of aroma compounds in Baijiu, so this method has not been described before, but we have now supplemented it on line 499 to 504. In detail, we introduce this method and its practical application "Therefore, some researchers have come up with other approaches, such as TDS (temporal dominance of sensations). The latest TDS claims to record multiple sensory attributes simultaneously over time and the researches based on TDS and TI have been carried out widely. In future, it will provide a new impetus and lay a foundation for sensory evaluation on Baijiu." was added at the end of section 5.2. Thank you again for your review.

Point 7: I'd recommend the authors to expand the sensory section as it is rather lacking while the other sections is already fine.

Response 7: We are very grateful to your comments for the manuscript. Thank you for your patient review and high evaluation of our work. We have made relevant additions to sensory section and described the aroma compounds in each category by category. It include esters, alcohols, acids, aldehydes, ketones, acetals, heterocyclic, phenolic, sulfur compounds and nitrogen compounds. We have described the characteristics and representative aroma compounds on the basis of category in detail and refined this part on line 761 to 790."Among them, esters contribute the aroma of fruit, flower and sweet. Alcohols main represent the aroma of fruit and fatty. Acids show the aroma of sour. Aldehydes and ketones show the aroma of herbal and buttery. Acetals mainly show the aroma of grass, fruit and sweet. Heterocyclic compounds mainly show nutty aroma and sweet aroma. Phenolic compounds mainly exhibit smoky aroma. Of note, the reason why sesame flavor Baijiu differs from other flavor types of Baijiu is presumed to be attributed to its sulfur compounds and nitrogen compounds (as shown in Figure 4). Sulfur compounds mainly expressed roasted incense, scorched incense, coffee incense, and nitrogen compounds mainly present nutty aroma, sweet aroma and roasted aroma. Therefore, sulfur compounds and nitrogen compounds are supposed to be the characteristic aroma compounds of sesame flavor Baijiu." was reversed as "Among them, esters contribute the aroma of fruit, flower and sweet. Although sesame flavor Baijiu is characterized by roasted sesame flavor, which is significantly different from other flavor Baijiu, its main aroma compounds is still esters. Esters play an important role in the aroma profile for sesame flavor Baijiu and present fruity aroma. Of note, ethyl hexanoate (FD factor = 4096, OAV = 2691) , ethyl butanoate (FD factor = 2048, OAV = 447), ethyl pentanoate (FD factor = 2048, OAV = 1019), ethyl octanoate (FD factor = 25, OAV = 782) are the main aroma compounds of esters in sesame flavor Baijiu. Alcohols are the precursors of esters, it main represents the aroma of fruit and fatty. For instance, 1-propanol (FD factor = 100, OAV = 6), 2-methyl-1-propanol (FD factor = 8, OAV = 4) are the common alcohols in sesame flavor Baijiu. Acids contribute sour taste for flavor profile of sesame flavor Baijiu and affect the taste and the aftertaste of sesame flavor Baijiu. The lack of acids in sesame flavor Baijiu is the main reason for the poor aftertaste. butanoic acid (FD factor = 1024, OAV = 57), pentanoic acid (FD factor = 256, OAV = 46), and hexanoic acid (FD factor = 2048, OAV = 35) are deemed as the main acids in sesame flavor Baijiu. Aldehydes and ketones are also important aroma compounds in sesame flavor Baijiu and they show the aroma of herbal and buttery. For instance, 3-methylbutanal (FD factor = 8, OAV = 2403 and β-damascenone (OAV = 116) are recognized as important aroma compounds in sesame flavor Baijiu. Acetals primary show the aroma of grass, fruit and sweet, such as 2-furaldehyde diethyl acetal (FD factor = 4). Heterocyclic compounds mainly contribute nutty aroma and sweet aroma for sesame flavor Baijiu, such as 2-acetyl furan (FD factor = 25) and 2-acetyl-5-methyl furan (FD factor = 5). Phenolic compounds mainly exhibit smoky aroma, such as phenol (FD factor = 4), 4-methylphenol (FD factor = 1, OAV = 43) and 4-ethyl-2-methoxyphenol (FD factor = 4, OAV = 2). Of note, the reason why sesame flavor Baijiu differs from other flavor types of Baijiu is presumed to be attributed to its sulfur compounds and nitrogen compounds (as shown in Figure 4). Sulfur compounds mainly expressed roasted incense, scorched incense, coffee incense, and nitrogen compounds mainly present nutty aroma, sweet aroma and roasted aroma. Therefore, sulfur compounds and nitrogen compounds are supposed to be the characteristic aroma compounds of sesame flavor Baijiu." Many thanks again for your review.

We tried our best to improve the manuscript and made some other changes in the manuscript. These changes will not influence the content and framework of the paper. And here we did not list the changes but marked by red in revised version. We appreciate the reviewers’ warm work earnestly, and hope that the correction will meet with approval. If there are any problems about our paper, please do not hesitate to let us know.

Dear editor,
We will revise the manuscript as soon as possible.
Thank you for your letter

Thank you and best regards,
Yours sincerely,
Dongrui Zhao

Key Laboratory of Brewing Molecular Engineering of China Light Industry
Beijing Laboratory for Food Quality and Safety
Beijing Technology and Business University, Beijing, China 100048
Phone: 86-10-68984890. Fax: 86-10-68984890
E-mail: zdrui6789@sina.com

Reviewer 3 Report

I have reviewed the manuscript entitled "Uncover the flavor code of roasted sesame for sesame flavor Baijiu: Advance on the revelation of aroma compounds in sesame flavor baijiu by means of modern separation technology and molecular sensory evaluation, by Chen et al. This review paper summarizes the revelation of aroma compounds in sesame flavor Baijiu. The paper is well presented and easy to read. The introduction provides a good, generalized background of the topic that quickly gives the reader an appreciation of the wide range of applications for Baijiu flavor. The literature cited is relevant to the study. I think the paper could prove to be very interesting and useful to very large researchers.

Detailed remarks about the text are as follows:

line 38: brewed was changed as fermented

line 510: Add a reference here: …. characteristics for aroma compounds in samples (Kesen et al., 2018).

Kesen, S.; Amanpour, A.; Tsouli Sarhir, S.; Sevindik, O.; Guclu, G.; Kelebek, H.; Selli, S. Characterization of Aroma-Active Compounds in Seed Extract of Black Cumin (Nigella sativa L.) by Aroma Extract Dilution Analysis. Foods 20187, 98. https://doi.org/10.3390/foods7070098

Improve the resolution of Figures.

Author Response

Response to Reviewer 3 Comments

Thank you very much for your comments concerning our manuscript entitled “Uncover the flavor code of roasted sesame for sesame flavor Baijiu: Advance on the revelation of aroma compounds in sesame flavor baijiu by means of modern separation technology and molecular sensory evaluation” (ID: foods-1611064). Those comments are all valuable and very helpful for revising and improving our paper. We have studied comments carefully and have made correction which we hope meet with approval. Revised portions were marked by red in "Tracked Changes" mode in our revised version. Other changes are also highlighted in red. The main corrections in the paper and the responds to the reviewer’s comments are as following:

Point 1: I have reviewed the manuscript entitled "Uncover the flavor code of roasted sesame for sesame flavor Baijiu: Advance on the revelation of aroma compounds in sesame flavor baijiu by means of modern separation technology and molecular sensory evaluation, by Chen et al. This review paper summarizes the revelation of aroma compounds in sesame flavor Baijiu. The paper is well presented and easy to read. The introduction provides a good, generalized background of the topic that quickly gives the reader an appreciation of the wide range of applications for Baijiu flavor. The literature cited is relevant to the study. I think the paper could prove to be very interesting and useful to very large researchers.

Response 1: We are very grateful to your comments for the manuscript. Thank you for your patient review and high evaluation of our work. It is also our honor to contribute to the development of sesame flavor Baijiu.

Point 2: line 38: brewed was changed as fermented.

Response 2: Many thanks for your fair advice. According to your suggestion, we have revised the original text and changed the word. Thank you again for your careful review.

Point 3: line 510: Add a reference here: …. characteristics for aroma compounds in samples (Kesen et al., 2018).

Kesen, S.; Amanpour, A.; Tsouli Sarhir, S.; Sevindik, O.; Guclu, G.; Kelebek, H.; Selli, S. Characterization of Aroma-Active Compounds in Seed Extract of Black Cumin (Nigella sativa L.) by Aroma Extract Dilution Analysis. Foods 2018, 7, 98. https://doi.org/10.3390/foods7070098.

Response 3: Thank you for your kind suggestion. This reference does lend support to the analysis and systematic review of our paper. Moreover, it is really helpful to our future research. We have quoted this reference according to your suggestion on line 518. Many thanks again for your review.

Point 4: Improve the resolution of Figures.

Response 4: Special thanks to you for your good comments and suggestions. According to your suggestion, we have improved the resolution of Figures. You can see the new pictures on line 77, 108, 756 and 788. Thank you again for your careful review.

We tried our best to improve the manuscript and made some other changes in the manuscript. These changes will not influence the content and framework of the paper. And here we did not list the changes but marked by red in revised version. We appreciate the reviewers’ warm work earnestly, and hope that the correction will meet with approval. If there are any problems about our paper, please do not hesitate to let us know.

Dear editor,

We will revise the manuscript as soon as possible.

Thank you for your letter

Thank you and best regards,

Yours sincerely,

Dongrui Zhao

Key Laboratory of Brewing Molecular Engineering of China Light Industry

Beijing Laboratory for Food Quality and Safety

Beijing Technology and Business University, Beijing, China 100048

Phone: 86-10-68984890. Fax: 86-10-68984890

E-mail: zdrui6789@sina.com
